# DNA Sequencing of CD138 Cell Population Reveals TP53 and RAS-MAPK Mutations in Multiple Myeloma at Diagnosis

**DOI:** 10.3390/cancers16020358

**Published:** 2024-01-14

**Authors:** Mihaela Dragomir, Onda-Tabita Călugăru, Bogdan Popescu, Cerasela Jardan, Dumitru Jardan, Monica Popescu, Silvia Aposteanu, Sorina Bădeliță, Gabriela Nedelcu, Cătălin Șerban, Codruța Popa, Tatiana Vassu-Dimov, Daniel Coriu

**Affiliations:** 1Faculty of Biology, University of Bucharest, 030018 Bucharest, Romania; dragomir_mihaela28@yahoo.com (M.D.); tatiana.vassu@bio.unibuc.ro (T.V.-D.); 2Fundeni Clinical Institute, 022328 Bucharest, Romania; cerasela.jardan@umfcd.ro (C.J.); monica2982003@yahoo.com (M.P.); silviaaposteanu@gmail.com (S.A.); sorinabadelita@gmail.com (S.B.); gabi.nedelcu12@gmail.com (G.N.); serbannicucatalin@yahoo.com (C.Ș.); delia.popa@umfcd.ro (C.P.); daniel.coriu@umfcd.ro (D.C.); 3Hematology Department, “Carol Davila” University of Medicine and Pharmacy, 050474 Bucharest, Romania; bogdan.popescu2@drd.umfcd.ro; 4Molecular Biology Laboratory, Medlife Bucharest, 010093 Bucharest, Romania; d.jardan@gmail.com

**Keywords:** multiple myeloma, NGS, plasma cells

## Abstract

**Simple Summary:**

Multiple myeloma is a hematologic neoplasm caused by abnormalities of plasma cells. Although cytogenetic-based risk stratifications are in clinical use, the role of genomic mutations in prognosis assessment is incompletely understood. Moreover, the sample source for molecular testing is a matter of debate, with recent data pointing out that seeking for mutations in plasma cells would increase the yield. We hereby compared both cytogenetic abnormalities and mutations detected by NGS in bulk bone marrow samples and CD138+ enriched plasma cells from patients diagnosed with multiple myeloma. We performed a FISH analysis, and then we used NGS to assess mutations in NRAS, KRAS, BRAF, and TP53 in the plasma cells and bulk bone marrow samples. The NGS data showed that the sequencing of the CD138+ plasma cells provided a more sensitive approach by identifying more variants in BRAF, KRAS, and TP53 compared to bulk marrow sequencing.

**Abstract:**

Multiple myeloma is a hematologic neoplasm caused by abnormal proliferation of plasma cells. Sequencing studies suggest that plasma cell disorders are caused by both cytogenetic abnormalities and oncogene mutations. Therefore, it is necessary to detect molecular abnormalities to improve the diagnosis and management of MM. The main purpose of this study is to determine whether NGS, in addition to cytogenetics, can influence risk stratification and management. Additionally, we aim to establish whether mutational analysis of the CD138 cell population is a suitable option for the characterization of MM compared to the bulk population. Following the separation of the plasma cells harvested from 35 patients newly diagnosed with MM, we performed a FISH analysis to detect the most common chromosomal abnormalities. Consecutively, we used NGS to evaluate NRAS, KRAS, BRAF, and TP53 mutations in plasma cell populations and in bone marrow samples. NGS data showed that sequencing CD138 cells provides a more sensitive approach. We identified several variants in BRAF, KRAS, and TP53 that were not previously associated with MM. Considering that the presence of somatic mutations could influence risk stratification and therapeutic approaches of patients with MM, sensitive detection of these mutations at diagnosis is essential for optimal management of MM.

## 1. Introduction

Multiple myeloma (MM) is a hematologic neoplasm with a high mortality rate that occurs relatively early, even in low-risk patients. According to the American Cancer Society, the 5-year Relative Survival Rate is 58% for all seer stages combined [1]. MM develops as a result of an abnormal transformation and proliferation of bone marrow plasma cells due to various genetic aberrations [2]. The primary events known to occur at the onset of the disease include chromosomal abnormalities that stratify patients in risk groups: standard risk (trisomies, t(11;14), t(6;14), hyperdiploidy, and normal karyotype), intermediate risk (t(4;14) and gain 1q), and high risk (del17p13, t(14;16), and t(14;20)) [3,4]. Evidence from MM sequencing studies suggests that plasma cell disorders are caused not only by cytogenetic abnormalities but also by mutations in major oncogenes and tumor-suppressor genes. Such molecular events typically occur late in the disease evolution [3]. Several somatic mutations affect pathways involved in RNA processing, protein translation, and protein response. These studies found that the most common mutated genes involved in the pathogenesis of MM are NRAS, KRAS, DIS3, BRAF, and TP53. Considering the MM potential to evolve clinically into acute leukemias, it is not unexpected that these genes are also involved in the malignant transformation of other hematopoietic lineages. The importance of these molecular findings is highlighted by the potential to revise the risk and prognosis of the disease but also by the possibility to selectively target them with personalized therapies, including MAPK pathway-targeted inhibitors [5]. The limitations of cytogenetic techniques can lead to erroneous results and inadequate disease management and risk stratification. Therefore, it is necessary to detect molecular abnormalities to improve the diagnosis and management of MM.

The aim of this study is to determine whether NGS can provide complementary findings in addition to cytogenetics to better evaluate risk stratification and therapeutic management in MM. Furthermore, we aim to provide a better understanding of the early pathogenic molecular mechanisms of the disease and to establish whether sequencing of the CD138 cell population is a more suitable option than bulk population to identify somatic mutations in MM.

## 2. Materials and Methods

### 2.1. Patient Groups

This study was performed in the Molecular Biology Laboratory of the Hematology Department at the Fundeni Clinical Institute, Bucharest, using bone marrow aspirates from a cohort of 51 patients diagnosed with multiple myeloma between 2019 and 2022. Samples were harvested only at diagnosis, reflecting the molecular landscape of patients naïve to therapy. For the diagnosis of MM, patients underwent clinical examination and cytogenetic and immunochemical analysis of the bone marrow. This study was conducted in compliance to the principles of the Helsinki Declaration and Institutional Review Board (IRB) and approved by the ethics committee of the Fundeni Clinical Institute. Prior to inclusion in the study, informed written consent was obtained from all patients for the scientific use of their data. 

### 2.2. CD138 Cells Separation

We separated plasma cells (CD138+) from samples harvested from 35 patients in the cohort. Initially, the counts of the CD138+ cells in the bone marrow aspirates were estimated via flow cytometry (Figure 1A). Next, plasma cells (CD138+) were isolated from heparinized bone marrow aspirate (5 mL) using the MACS protocol (Miltenyi Biotec, Bergisch Gladbach, Germany), according to the manufacturer’s protocol. Bone marrow mononuclear cells were separated in a density gradient using the Ficoll–Paque. The CD138+ plasma cells were magnetically labeled with CD138 MicroBeads and loaded onto a MACS^®^ column placed in the magnetic field of a MACS Separator. Magnetic labeling was performed on at least 1 × 10^6^ cells per sample. In samples subjected to downstream FISH analysis, the enriched plasma cells were treated with KCl and a fixative solution (methanol and acetic acid 3:1).

### 2.3. Fluorescence In Situ Hybridization Analysis

The most frequent chromosomal abnormalities described in MM were assessed by interphase FISH: TP53/D17Z1 for detecting del17p; CKS1B(1q21.3)/CDKN2C(1p32.3) for chromosome 1 abnormalities; dual-color translocation probes MAF(16q23)/IGH(14q32.3) for t(14;16); and FGFR3(4p16)/IGH(14q32.3) for detecting translocation t(4;14) (CytoCell^®^, Oxford Gene Technology, Kidlington, UK). The samples were denatured at 75 °C using Hychrome (Euroclone, Pero, Italy) and hybridized at 37 °C for 16 h. A total of 100 nuclei were counted to determine cut-off values for the FISH analysis. The upper limits of the normal cut-off were set at 10% for the whole panel. An Olympus BX41 fluorescence microscope was used for imaging and the results were reported according to ISCN 2020 (Figure 1B).

### 2.4. DNA Extraction

Total DNA was isolated from either bulk BM aspirates or CD138+ enriched plasma cells. For CD138 cells, the fixative solution was replaced with 200 μL PBS after centrifugation of the samples (10,600 rpm, 10 min, 4 °C) and extraction of the supernatant. Subsequently, DNA extraction (PureLink™ Genomic DNA Mini Kit, Invitrogen™, Waltham, MA, USA) was conducted according to the manufacturer’s instructions for both CD138 cells and BM samples. DNA elution was performed in a final volume of 50 μL. The DNA concentration (3.1–127.3 ng/µL) and the purity (A260/A230, 1.41–1.92; A260/A280, 1.46–2.25) were measured by spectrophotometry (NanoDrop™1000, Thermo Scientific™, Waltham, MA, USA).

### 2.5. Next-Generation Sequencing and Data Analysis

The libraries for the target regions of BRAF, HRAS, NRAS, and KRAS genes were constructed using Primer-BLAST software. The target regions and their respective primers are detailed in Appendix A. For TP53 mutations we used Accel-Amplicon Comprehensive TP53 Panel (AL-53048) (Swift BioSciences, Ann Arbor, MI, USA). The purification of the amplicons was performed using ProNex^®^ Size-Selective Purification System (Promega, Madison, WI, USA). Samples were pooled in an equimolar ratio. The quality and quantity of the samples were measured using a Qubit™ 4 Fluorometer (Thermo Fisher Scientific, Waltham, MA, USA). Multiplex sequencing was performed with a 500-cycle double indexed paired-end run on a MiSeq sequencer (Illumina, San Diego, CA, USA) at a median depth of 330× and IQR = 1.462×. For demultiplexing, we used a CASAVA 1.8 (Illumina, San Diego, CA, USA). The resulting FASTQ files were aligned to the GRCh38 reference genome using DNASTAR Lasergene 17.2 software and visualized with IGV 2.16.0 software. To annotate the identified variants, we used the Ensembl Database and HGVS nomenclature. All sequences were manually analyzed using ClinVar and COSMIC Databases, according to the American Society of Clinical Oncology and College of American Pathologists Guidelines [6,7].

### 2.6. Bioinformatic Tools

Variant interpretation was performed using the following databases: the Catalogue of Somatic Mutations in Cancer—COSMIC (Sanger Institute, Saffron Walden, UK), ClinVar Database (National Library of Medicine, Bethesda, MD, USA), and Genome Aggregation Database (gnomAD). For pathogenic/likely pathogenic mutations that are not associated with MM (BRAF-V600G, KRAS-G13D, TP53-L114*), the 3D design of the resulting proteins was obtained using Swiss-Model, Biozentrum. The ridgeline plot, box plot, and C-Net plot were performed using http://www.bioinformatics.com/srplot (accessed on 8 August 2023). 

## 3. Results

### 3.1. Demographic and Clinical Profile of Study Subjects

Baseline characteristics of the patients and treatment particularities are described in Appendix A. The median age of the patients in the cohort was 61 years and the male-to-female ratio was 27:24. Most of the subjects (90.19%) carried at least one significant cytogenetic aberration (Figure 2). The most frequent cytogenetic abnormality was represented by del17p13 (35.29%). According to the ISS classification, 50.98% of patients were included in the ISS-III class. In terms of risk stratification (based on identified cytogenetics abnormalities), standard, intermediate, and high-risk subjects account for 41.17%, 15.68%, and 43.13%, respectively [4,8]. Also, 11 patients (21.56%) were refractory to the last received treatment line, either induction or maintenance therapy (Appendix A). Death occurred in the case of six patients due to disease progression.

### 3.2. Risk Stratification According to Cytogenetics

Cytogenetic abnormalities were found in most subjects (>90%), thus allowing their inclusion in risk groups depending on the chromosomal alterations detected [4]. Patients who presented t(11;14) (1.96%) and normal karyotype (9.81%) were included in the standard risk group. Also, the cases in which t(4;14) (17.64%) and gain 1q (21.56%) were detected were classified as intermediate risk, and those who presented del17p (35.29%) and t(14;16) (7.84%) were considered to present high risk. Thus, depending on these groups, most patients exhibited high risk (43.13%) and standard risk (41.17%). Only a small percentage of the study group was classified as intermediate risk (15.68%). Due to the high percentage of del17p, gain 1q, and t(4;14) in patients refractory to the last line of treatment, most of them were classified as high risk and intermediate risk (54.54%). An interesting observation is that within the refractory group, cytogenetics performed on bulk BM samples (54.5%) did not show any genetic abnormality and therefore, those patients were included in the standard risk group. On the other hand, cytogenetics in the enriched plasma cells (45.5%) revealed del17p, gain 1q, and t(4;14), which required classifying these patients as high and intermediate risk.

### 3.3. Next-Generation Sequencing and Data Analysis

#### 3.3.1. Variant Interpretation

Analysis of sequencing data from both groups (CD138 in BM vs bulk cells in BM) revealed the presence of mutations predominantly in HRAS (37%), followed by TP53 gene (33%), KRAS (18%), NRAS (12%), and BRAF (6%). However, mutations in HRAS were not considered pathogenic/potentially pathogenic mutations, all being synonymous variants (H27=, L79=, V81=). At the same time, in 3.91% of the patients, the association of several pathogenic or potentially pathogenic mutations was observed (Figure 2). Of the total mutations identified, 23.52% are identified by the ClinVar database as pathogenic or likely pathogenic but only half of them are associated with MM. Also, two variants identified in TP53 (L130F) and KRAS (G13D) genes presented conflicting interpretations of pathogenicity. In terms of mutation type, missense (78.43%), intronic (7.84%), nonsense (1.96%), coding silent (1.96%), and frameshift mutations (1.96%) are distinguished (Table 1). An interesting observation is the intra-patient association of several pathogenic/potentially pathogenic mutations in patients who, prior to sequencing, were included in the standard/intermediate risk group (Figure 2) [4,8]. Such cases totaled 11.76%, and harbored gain 1q21, t(4;14), and no chromosomal abnormalities but associated pathogenic/potentially pathogenic mutations in NRAS, KRAS, BRAF, or TP53. We observed that 63.63% of patients without del17p13 harbor mutations in the TP53 gene and 3.92% of patients with normal karyotype have pathogenic/potentially pathogenic mutations in at least one of the analyzed genes. 

Most mutations were identified in the CD138 population, with bulk cells harboring only one certainly pathogenic variant in the BRAF gene.

Furthermore, within the patient’s refractory to the last line of treatment, we identified several mutations in NRAS, KRAS, and BRAF genes (36.4%). These mutations co-occurred with chromosomal aberrations, such as del17p, gain1q, and t(4;14) (Appendix A). Similar to cytogenetic analysis, performing NGS on bulk BM samples did not identify any pathogenic mutations, while all the MAPK pathway genes mutations were found in the enriched CD138 plasma cells.

#### 3.3.2. Mutational Burden

In agreement with other similar studies, we described the mutational burden using cancer clonal fraction (CCF) as minor (<20% CCF), prominent (20–60% CCF), major (≥60% CCF), and clonal CCF ≥ 90% (Figure 3) [9]. A fraction of patients (21.56%) had associated low-burden and high-burden mutations, highlighting the MM tumoral heterogeneity and the complexity of the clonal architecture. Almost two-thirds of the samples had a prominent CCF frequency. Given that pathogenic or potentially pathogenic mutations can exhibit a wide range of mutational burden frequency (3–79%), this highlights the importance of detecting variants at very low frequencies in patients diagnosed at disease onset. We observed that the gene with the highest variance in the mutational burden was TP53 (Figure 3). The most frequently identified variant is P72R in the TP53 gene (56.86%) with a high degree of heterogeneity (42–100%). After querying this variant in the Genome Aggregation Database (gnomAD) and based on the allelic frequency, this variant is considered to date to be a polymorphism [10].

### 3.4. In Silico Modeling of Mutated Proteins

Given that information regarding the molecular impact of our identified mutations in MM pathogeny is rather scarce, we performed a bioinformatic analysis to identify possible interaction pathways of the target genes (Figure 4).

We next performed in silico modeling for the potentially pathogenic variants (BRAF-V600G, KRAS-G13D, and TP53-L114*) using Swiss-Model (Biozentrum, Vienna, Austria). In a 3D protein model of the V600G variant (BRAF gene), we observed conformational changes in the SSDD motif and the hydrophobic landscape of G600. Structural changes between I371 and D449 (included in the SSDD motif) are highlighted in Figure 5. These types of alterations of the SSDD motif and the V600 residue suggest a gain in function of the resulting protein. Analysis of KRAS G13D revealed a structural alteration of the Switch II pocket involving residues S65-M67, which also has the potential to activate the KRAS proto-oncogene (Figure 6). An in silico-generated model of the L114* variant of TP53 reveals a loss in function mutation type. L114* is located in the DBD domain (Figure 7). Since this variant induces a premature stop codon in the AA sequence, it will result in a truncated protein with abolished function and subsequent loss-of-function and impairment of the tumor suppressor ability of TP53.

## 4. Discussion

Mutations in TP53 or those constitutively activating the MAPK signaling pathway have been shown to confer a worse outcome in solid tumors and AML [11,12,13,14,15] (Figure 4). 

Although several guidelines attempt to standardize diagnosis, treatment, and risk stratification using cytogenetics and recommend NGS to track clonality in MM, the mutational landscape still remains insufficiently explored [4,8]. Longitudinal whole-exome sequencing studies in MM have shown that driver mutations in TP53 or RAS/MAPK mutations are enriched at relapse, although they are present as subclonal events at presentation, suggesting that clonal expansion drives disease progression and relapse [9]. In line with these findings, we observed a lower abundance or absence of RAS/RAF mutations in bulk BM cells compared to the plasma cell population at presentation. Therefore, we hypothesize that although such molecular changes are frequently absent in bulk marrow, they can still be identified at the onset of MM by analyzing the CD138 cell population and can define subclones that can drive progression. Considering that a significant percentage of patients with standard/intermediate risk can harbor pathogenic/potentially pathogenic mutations, the combination of plasma cell separation and sequencing the CD138-positive fraction is a reliable option to detect somatic mutation at diagnosis in MM. By enhancing the sensitivity of detecting molecular variants, the analysis of the CD138 population can provide a better overall picture of the clonal evolution, reassess the risk stratification, and inform on potential clinical benefits of personalized therapies. Although the implications of BRAF mutations in the pathogenesis of solid cancers (especially melanoma) are well established and BRAF inhibitors are already in clinical use for patients with melanoma and other malignancies, the role of BRAF in hematological malignancies is still underexplored [16,17,18]. In our study, we identified three different mutations in the BRAF gene. According to the ClinVar Database, p.V600G (c.1799T>G) and p.K601E (c.1801A>G) are characterized as pathogenic, but they have not been associated yet with MM. Both mutations correspond to the third conserved region of the BRAF gene (CR3), that harbors the kinase domain, activation segment (AS), and phosphorylation sites [19]. V600G (previously known as V599G) lies within the activation segment (AS) of the kinase domain of the B-Raf protein, adjacent to the conserved DFG motif (Figure 5). 

In the inactive conformation, V600 is buried in a hydrophobic pocket made by residues from the N-terminal region and the AS. Destabilization of these interactions can induce kinase activation and phosphorylation within the S446SDD449 motif which is responsible for the negative charge of the N-terminal region (Figure 5) [20,21,22]. It has been shown that all the V600 pathogenic variants belong to the high-activity mutant class [23,24,25]. Although most BRAF variants require interaction with the RAS for phosphorylation and activation, V600 mutants can overcome this demand by mimicking phosphorylation and constitutively activating downstream signaling [26]. Moreover, in vitro studies show that V600G represents a gain of function variant associated with increased phosphorylation of the MEK and ERK and increased cell proliferation and viability compared to BRAF^WT^ but less strongly activating than the V600E mutation [20,27]. Phosphorylation of the T599VKS602 motif is supposed to disrupt the normal interactions and initiate the subsequent phosphotransferase reaction. Alternatively, AS phosphorylation might contribute to conformational changes in the kinase domain, leading to catalysis and allosteric activation [19]. As we observed with the BRAF V600G mutation, the identified substitution produces effects on the structure of the resultant protein characterized by modifications in protein structure at a variable distance from the causal mutation. 

Even if it is not associated with MM (according to the ClinVar Database), K601E occurs in at least 1% of melanoma and adenocarcinoma [28,29,30]. It is associated with high kinase activity by its interaction with the phosphate-binding loop, and it could also increase the activity of the downstream MAPK pathway [31]. Compared with V600E, it is more sensitive to trametinib than vemurafenib and shows better clinical outcomes in melanoma and small-cell lung cancer (NSCLC) [30,32,33]. Additionally, V600G showed a good response to dabrafenib in thyroid carcinomas and melanoma [34]. Although the single or combined BRAF/MEK inhibitors (encorafenib, binimetinib, vemurafenib, dabrafenib, and trametinib) showed efficacy in refractory MM in recent phase 2 clinical trials, this response was maintained for a short period of time [35,36,37]. However, these studies included only small groups of patients, and therefore, larger cohorts are needed to be able to draw accurate conclusions. Moreover, an important caveat of the BRAF inhibitors monotherapies is the paradoxical activation of the MAPK pathway, according to which, a RAS^mut^ background contraindicates the administration of BRAF inhibitors due to tumor potentiation [38,39].

Although, according to ClinVar Database, the KRAS G13D variant presents conflicting interpretations of pathogenicity and it is not associated with MM, in vitro studies showed an increased activity of proteasome in MM cell lines expressing KRAS G13D [40,41]. To better understand the pathogenic potential of the KRAS G13D, we performed a 3D modeling of the variant. This variant is localized in P-loop region (10–14), where any modification of the G-residues (G12; G13) induces changes in KRAS protein structure and its activation [42]. Analysis of the KRAS in silico model reveals that even the G13D variant is established in the P-loop region; the conformational changes in the resultant protein occur downstream of this region in residues 65–67 of the Switch II pocket (58–72) (Figure 6). 

It has been shown that 37% of the KRAS mutations are localized in this region and contribute to the conformation of the binding interface for effector proteins [43]. Also, any structural modifications of the P-loop could lead to recruiting other proteins through the Switch II pocket. Because of the structural similarities between the G13D and WT form, it was considered that the changes in KRAS dynamics occur in an allosteric manner and that the changes produced by the mutation produce effects in distant regions [43]. However, the G13D variant modifies not only the P-loop region but also leads to structural effects in the SII pocket. Although de novo KRAS or NRAS mutations were reported in MM patients at disease relapse, our findings indicate that it is possible to identify them even at diagnosis, if sequencing is performed on the CD138 population from BM [42]. This is particularly relevant in the context of therapies aimed at suppressing the RAS/MAPK signaling MEK inhibitors already in clinical trials for MM in the setting of RAS^mut^ background [44,45,46]. Although the efficacy, tolerability, and resistance to MEK inhibitors need to be established clinically, novel pan-RAS molecules that hold the promise to inhibit signaling downstream of all mutant and wild-type RAS isoforms are currently in preclinical development [44,47,48,49,50]. 

To date, one of the most important factors used in MM risk stratification is del17p, a cytogenetic aberration associated with poor outcomes, even in the absence of TP53 mutations. Recent studies showed that the coexistence of del17p and TP53 mutants (“double hit”) indicates a very high risk (above del17p alone) [51]. Because both del17p and TP53 mutations can be acquired during the evolution of MM, they need to be monitored throughout the course of treatment [51]. The most common variant identified in our study is the P72R polymorphism in the TP53 gene. This variant, although benign according to most databases, it presents controversies in terms of pathogeny. The P72R polymorphism is most common in higher latitudes and colder climates and, according to the gnomAD Database, it has the highest allelic frequency in the European population (0.7366) [10]. However, a study involving mouse models of ovarian cancer demonstrated that when associated with pathogenic variants of TP53, the R72 SNP promotes a higher growth rate of the TP53 missense mutants and accelerates cell proliferation of the p53 common target mutants. In contrast, the P72 SNP significantly suppresses cell growth and improves overall survival. Although these effects may be common in many missense p53 mutant proteins, the impact of the P72R SNP on disease severity in MM is a topic that requires further investigation [52]. 

Since the TP53 gene is a major tumor suppressor, most of its pathogenic mutations lead to the abolition of the normal gene function. These loss-of-function mutations enable the activation of proto-oncogenes, promote tumor proliferation, and reduce the apoptosis capacity of malignant clones. The most extensive functional domain of TP53 is represented by the DBD (DNA-Binding-Domain), whose major role is to stabilize the protein-DNA complex and initiate transcription [53]. Due to the instability of the DBD domain, most inactivating mutations are located at this level (hotspots). L114* is located in the DNA-Binding-Domain and was identified in the case of a single patient (CCF 39%) (Figure 7). 

L114* is a pathogenic, nonsense variant that introduces a premature stop codon into the normal amino acids sequence of TP53, leading to the truncation of the normal protein. Considering the location of the variant in the DBD field, this variant induces the loss of function of the p53 protein.

Although the detection of mutations in the TP53 gene and the RAS/MAPK pathway are particularly important, other molecular findings may be equally relevant to prognosis. Recent data emphasizes the role of the MYC gene, whose rearrangements have been shown to lead to progression and symptomatic disease, conferring MM patients who carry MYC rearrangements a poor prognosis [54]. In addition to detecting mutations at diagnosis, NGS may also prove to be a useful prognosis assessment tool if used as an MRD detection assay in addition to multiparameter flow cytometry [55]. 

Our study faces several limitations. This study is retrospective in nature and relies on banked primary samples harvested from patients diagnosed with MM. Due to limited sample availability, we were unable to perform intra-patient pairwise comparisons between sequencing bulk BM versus enriched CD138+ cells. To draw a definitive conclusion regarding the superiority of CD138 enrichment over bulk analysis, it is necessary to sequence both bulk and enriched cell populations harvested from the same patient at the same time. Nevertheless, the higher abundance of genetic lesions, both cytogenetic and molecular, identified in our CD138-enriched samples is suggestive of that conclusion. Due to our limited cohort size, definitive correlations between the presence of mutations and clinical responses could not be made. However, a significant number of patients who were refractory to the last line of therapy at the time of inclusion harbored at least one mutation or cytogenetic abnormality which was exclusively detected in the CD138+ fraction.

Furthermore, the technique of selecting CD138+ plasma cells from the bone marrow is laborious and requires an invasive procedure. Novel approaches, aimed at detecting genetic lesions within readily available circulating samples, such as circulating tumor cells or cell-free circulating tumor DNA, can overcome these limitations and are gaining traction, especially for MRD assessment [54,56,57]. 

## 5. Conclusions

The pathogenesis of multiple myeloma involves a constellation of genetic alterations that determine subsequent risk stratification and prognosis. There is an unmet need for the sensitive detection of TP53 and RAS-MAPK signaling mutations, for the purpose of both refining prognosis and assessing the clinical benefit of using targeted inhibitors. Until recently, chromosomal abnormalities were considered to be an early event in the disease evolution, but relying only on cytogenetic analysis can lead to an inadvertently simplistic image of the genetic background in MM. Bulk DNA sequencing from bone marrow aspirates did not historically reveal much information regarding the mutational status at diagnosis, hence the somatic mutations were considered late events in the disease evolution. However, cell enrichment techniques and sequencing of the CD138-positive population have the potential to identify pathogenic variants early and could influence therapeutic strategies. Ultimately, we have demonstrated that sequencing the CD138 positive fraction of plasma cells is a more appropriate approach than analysis of bulk bone marrow aspirate for the purpose of identifying mutations with prognostic significance and with potentially clinically addressable targeted therapies.

## Figures and Tables

**Figure 1 cancers-16-00358-f001:**
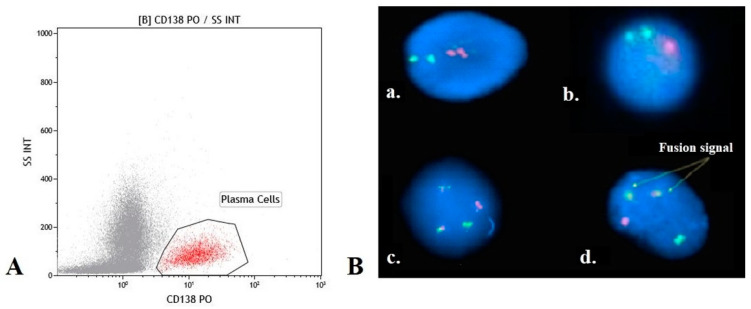
(**A**) Plasma cell number was determined by flow-cytometry; (**B**) Representative FISH analysis image: (**a**) normal pattern for del17p; (**b**) abnormal pattern for del17p; (**c**) normal pattern for t(4,16)/t(14,16); and (**d**) abnormal pattern for t(4,16)/t(14,16).

**Figure 2 cancers-16-00358-f002:**
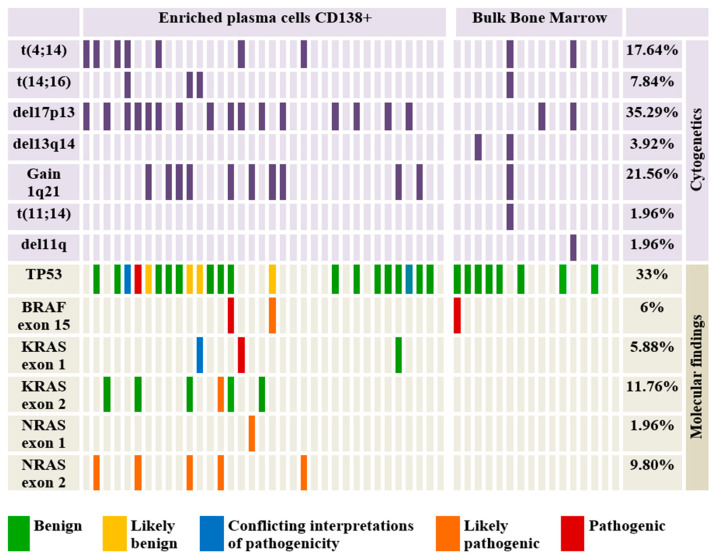
Oncoplot showing the distribution of the identified cytogenetic lesions and mutations identified by NGS and their putative pathogenicity.

**Figure 3 cancers-16-00358-f003:**
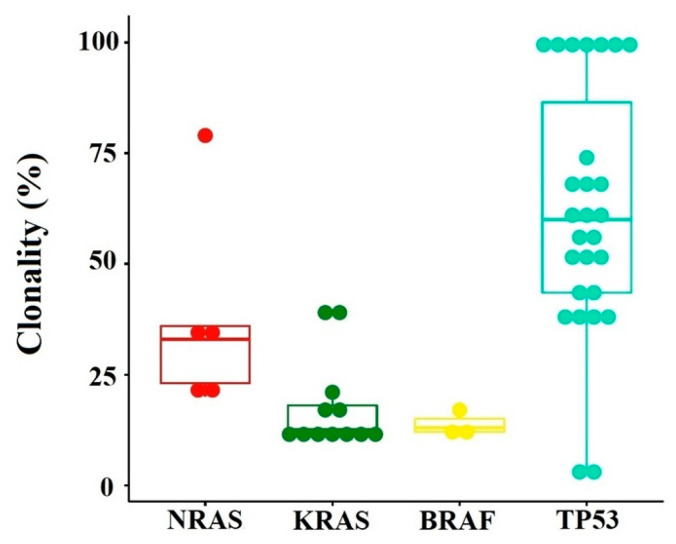
Mutational burden of variants identified in the patient cohor.

**Figure 4 cancers-16-00358-f004:**
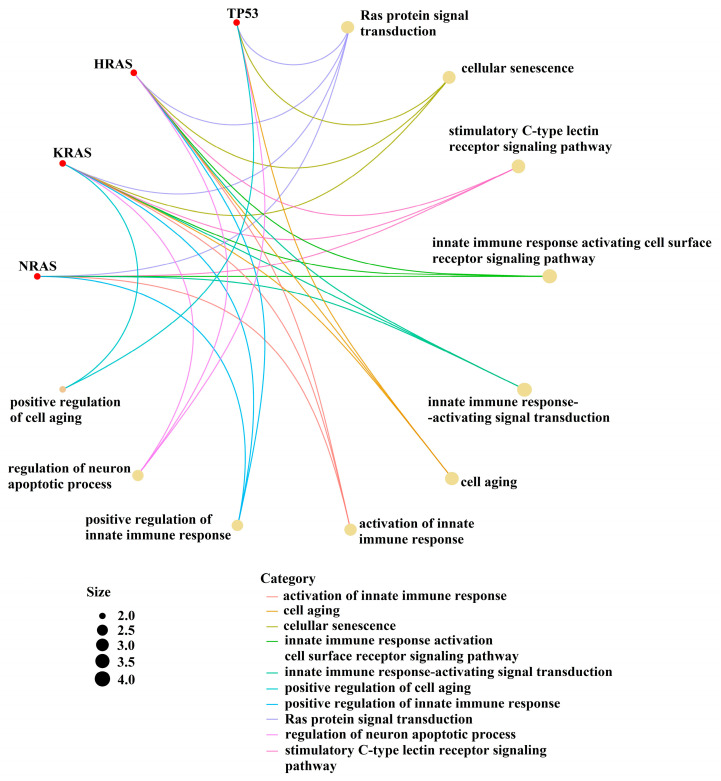
Interaction between the pathways of the target genes.

**Figure 5 cancers-16-00358-f005:**
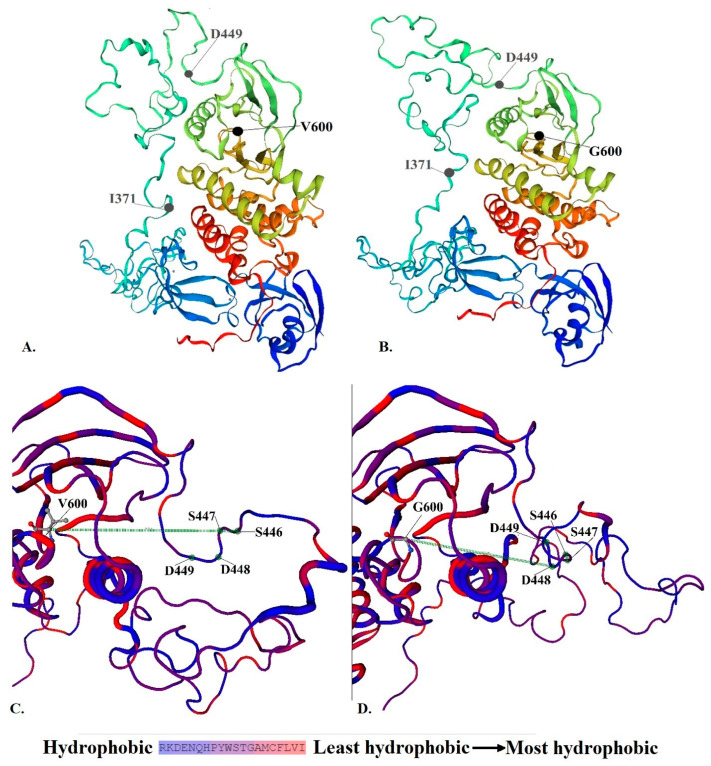
(**A**) Structure of B-Raf protein (**B**) Structure of V600G mutant (**C**) The normal conformation and hydrophobic status of B-Raf protein (**D**) The conformational changes in SSDD motif and hydrophobic landscape of V600G.

**Figure 6 cancers-16-00358-f006:**
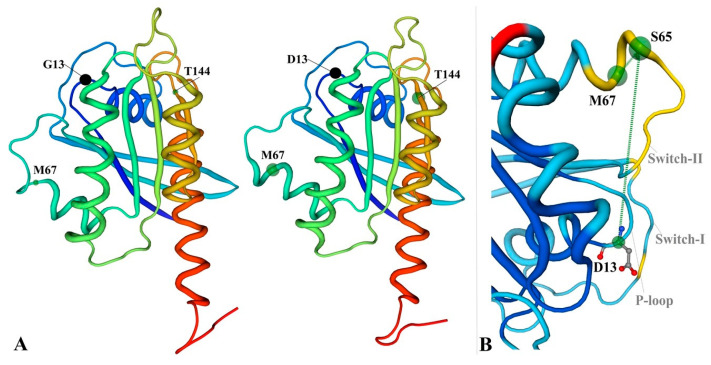
(**A**) Structure of KRAS protein and G13D mutant (**B**) The conformational changes in Switch-II pocket.

**Figure 7 cancers-16-00358-f007:**
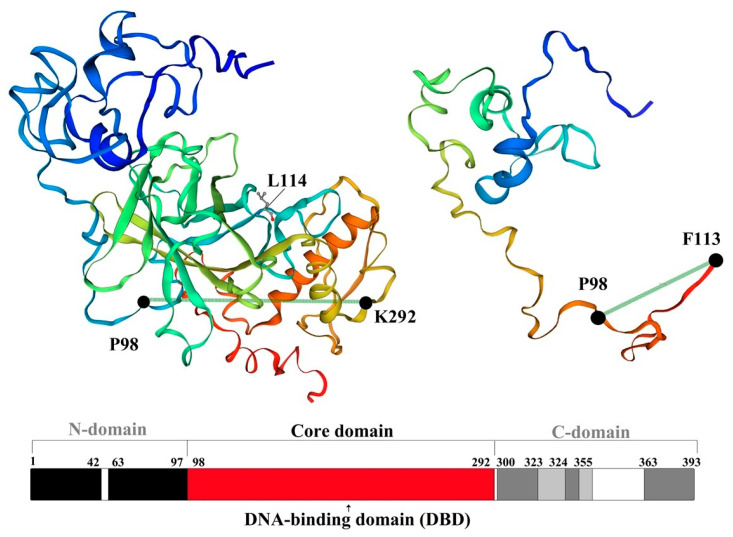
Structure of p53 protein and L114* mutant.

**Table 1 cancers-16-00358-t001:** Variants identified by NGS, allelic frequency, and significance.

Gene	Variant	AA Change	Mutation	Allelic Frequency	Significance (ClinVar)	No. of Patients
TP53	c.215C>G	p.P72R	Missense	42–100%	Benign	17
	c.250G>A	p.A84T	Missense	37–38%	Likely benign	3
	c.341T>A	p.L114 *	Nonsense	39%	Pathogenic	1
	c.388G>T	p.L130F	Missense		Conflicting interpretations of pathogenicity	1
	c.464C>T	p.T155I	Missense	55%	Uncertain significance	1
	c.639A>G	p.R213=	Coding silent	60%	Benign	1
	c.782+10C>T	-	Intronic	45%	Benign/Likely benign	1
	c.993+12T>C	-	Intronic	37–62%	Benign/Likely benign	3
BRAF	c.1900G>A	p.D594N	Missense	13%	Likely pathogenic	1
	c.1919T>A	p.V600G	Missense	11%	Pathogenic	1
	c.1921A>G	p.K601E	Missense	17%	Pathogenic	1
KRAS	c.35G>C	p.G12A	Missense	10%	Pathogenic	1
	c.38G>A	p.G13D	Missense	39%	Conflicting interpretations of pathogenicity	1
	c.183A>C	p.Q61H	Missense	17%	Likely pathogenic	1
	c.219G>A	p.R73=	Missense	11–21%	Benign	6
NRAS	c.34G>C	p.G12R	Missense	23%	Likely pathogenic	1
	c.181C>A	p.Q61K	Missense	15–79%	Likely pathogenic	4

The “*” it is an annotation for a codon stop.

## Data Availability

The data that support the findings of this study are available on request from the corresponding author.

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
