# Peer review of "DNA Sequencing of CD138 Cell Population Reveals TP53 and RAS-MAPK Mutations in Multiple Myeloma at Diagnosis"

_cancers, 2024, doi:10.3390/cancers16020358_

Round 1

Reviewer 1 Report

Comments and Suggestions for Authors

The authors screened for large cytogenetic aberrations (e.g. translocations) and small variants (e.g. point mutations) in a cohort of patients with multiple myeloma. They conclude “sequencing of the CD138 positive fraction of plasma cells is a more appropriate approach than analysis of bulk bone marrow aspirate”. Other aspects of the paper are rather descriptive.

Major point: 

A more appropriate approach would be to compare the diagnostic yield in bone marrow vs. anti-CD38 enriched plasma cells derived from the same patient (from one and the same bone marrow aspirate).

Minor points:

The description of the NGS data analysis is incomplete. Can this analysis detect indels/delins? As there are not germline control samples included, how were germline polymorphisms detected and excluded from analysis (VAF threshold, GnomAD, etc.)?

Figure 1A: What was the role of FACS in the anti-CD138 enrichment process? The authors mention that the MACS protocol was used for enrichment.

Figure 2: Peripheral blood? 

Supplemental Table 1: On what basis were the target regions selected? This seems to be only a very few exons per gene.

Lines 76-77: It is unclear to me how the authors “sorted” 106 cells from the samples. A usual workflow would be to enrich several hundreds to thousands of cells from several milliliters of bone marrow, perform FISH on several hundred cells and then analyze 100 high-quality interphases. Please describe your workflow in more detail and explain how to “handle” as little as 106 cells.

Line 78: What was the pellet used for?

Line 102: “The libraries for BRAF, HRAS, NRAS and KRAS genes were constructed…”. What proportion of all exons per gene? How many exons were not analyzed for the genes mentioned? Please state precisely what were your target regions for NGS.

Line 109: Please specify the range (e.g. interquartile) for the provided medium depth.

Line 131: What type of risk classification are the authors referring to?

Line 135: Please remove the “Figure 3 legend” from Figure 2.

Line 140: Based on what guidelines do you consider gain 1q and t(4;14) as intermediate risk?

Line 163: Please add some information on potential TP53 double hits (heterozygous deletion 17p AND mutated TP53).

Line 182: How does the frequency in the myeloma cohort compare to the GnomAD frequencies? Is there an enrichment in myeloma (supposedly somatic) as compared to GnomAD (supposedly germline)? What is the evidence that TP53 P72R is somatic in your MM cohort?

Line 206: 0.7366 – is this a percentage or a fractional abundance?

Line 337: What is the final evidence that NGS of CD138 is superior over NGS of bulk BM DNA? This relates to the major point stated above.

Comments on the Quality of English Language

none

Author Response

Thank you very much for your honest review. You make very good points and our manuscript can really benefit from them. I have attached the file with our answers.

Major point: 

A more appropriate approach would be to compare the diagnostic yield in bone marrow vs. anti-CD38 enriched plasma cells derived from the same patient (from one and the same bone marrow aspirate).

R: We acknowledge that the major limitation of our study is not performing a pairwise bulk marrow vs CD138+ plasma cells comparison within same patient samples. We agree that your suggestion is a more appropriate approach, but unfortunately we were heavily limited by patient sample availability. This is a retrospective study using bio-banked samples and BM and enriched plasma cells derived from the same patient were not available. We updated the the discussions paragraph to emphasize this limitation.

Minor points:

The description of the NGS data analysis is incomplete. Can this analysis detect indels/delins? As there are not germline control samples included, how were germline polymorphisms detected and excluded from analysis (VAF threshold, GnomAD, etc.)?

R: Yes, the analysis can detect indels/delins. The detected variants, including germline polymorphisms (for exemple TP53 – P72R) were reported according to American Society of Clinical Oncology and College of American Pathologists Guidelines. Although no germline control samples were included, we queried ClinVar Database to differentiate germline variants from somatic variants (lines 113-119).

Figure 1A: What was the role of FACS in the anti-CD138 enrichment process? The authors mention that the MACS protocol was used for enrichment.

R: We didn’t use FACS in the anti-CD138 enrichment process. MACS only was used for enrichment. We used flow cytometry to determine the number of CD138+ cells in BM samples prior to MACS enrichment. To avoid ambiguity, we updated the Figure 1A caption to avoid the misunderstanding: ”Figure 1. (A) Plasma cell number was determined by flow-cytometry”.

Figure 2: Peripheral blood? 

R: There was an error in the figure. We have corrected it.

Supplemental Table 1: On what basis were the target regions selected? This seems to be only a very few exons per gene.

R: We selected the target regions based on the mutational hotspots identified in public data repositories available on  cbioportal – Broad, Cancer Cell 2014 (Cancer Cell, 2014 Jan 13;25(1):91-101). We sequenced the hotspots most frequently mutated in that study – exon 15 for BRAF, and exons 1 and 2 (codons 12, 13, 60, 61) in NRAS and KRAS genes.

Lines 76-77: It is unclear to me how the authors “sorted” 106 cells from the samples. A usual workflow would be to enrich several hundreds to thousands of cells from several milliliters of bone marrow, perform FISH on several hundred cells and then analyze 100 high-quality interphases. Please describe your workflow in more detail and explain how to “handle” as little as 106 cells.

R: There is a typing error – the superscript was not shown. It was supposed to say 10^6 cells instead of “106 cells”. We corrected the methods section. We analyzed 1x10^6 cells using CD138 MicroBeads Protocol from Miltenyi Biotec.

Line 78: What was the pellet used for?

R: We have also changed the text paragraph in the methods section for a more accurate description of the method and to avoid the future misunderstandings:

“Plasma cells (CD138+) were isolated from heparinized bone-marrow aspirate (5ml) harvested from 35 patients, using the MACS protocol (Miltenyi Biotec), according to the manufacturer’s protocol. Bone marrow mononuclear cells were separated in density gradient using using Ficoll-Paque. The CD138+ plasma cells were magnetically labeled with CD138 MicroBeads and loaded onto a MACS® Column placed in the magnetic field of a MACS Separator. Magnetic labeling was performed on at least 1x10^6 cells per sample. Prior to FISH Analysis, the enriched plasma cells were treated with KCl and fixative solution.“

Line 102: “The libraries for BRAF, HRAS, NRAS and KRAS genes were constructed…”. What proportion of all exons per gene? How many exons were not analyzed for the genes mentioned? Please state precisely what were your target regions for NGS.

R: We have stated into the text manuscript that we didn’t sequence the whole gene, but only targeted regions: “The libraries for the target regions of BRAF, HRAS, NRAS and KRAS genes were constructed using Primer-BLAST software (Supplemental File)”. The targeted regions and the primers used are detailed in the supplemental file.

Line 109: Please specify the range (e.g. interquartile) for the provided medium depth.

The medium depth specified in the text (250 reads is the one recommended by American Society of Clinical Oncology and College of American Pathologists Guidelines. Our actual median depth is 330 reads (IQR=1,462)

Line 131: What type of risk classification are the authors referring to?

R: We have modified the line to: “In terms of risk stratification (based on identified cytogenetics abnormalities), standard, intermediate and high-risk subjects account for 41,17%, 15,68% and 43,13% respectively [4,8].”

Line 135: Please remove the “Figure 3 legend” from Figure 2.

R: We have removed it.

Line 140: Based on what guidelines do you consider gain 1q and t(4;14) as intermediate risk?

R: According to the 2022 update on diagnosis, risk stratification and management, published in American Journal of Hematology (https://doi.org/10.1002/ajh.26590), gain 1q and t(4;14) abnormalities are classified as intermediate risk.

Line 163: Please add some information on potential TP53 double hits (heterozygous deletion 17p AND mutated TP53).

R: We have detailed the coexistence and the prognostic impact of del17p and mutated TP53 in the Discussion section - lines 300-305 [To date, one of the most important factors used in MM risk stratification is del17p, a cytogenetic aberration associated with poor outcome, even in the absence of TP53 mutations. Recent studies showed that coexistence of del17p and TP53 mutants (“double hit”) indicate a very high-risk (above del17p alone). Because both del17p and TP53 mutations can be acquired during the evolution of MM, they need to be monitored throughout the course of treatment].

Line 182: How does the frequency in the myeloma cohort compare to the GnomAD frequencies? Is there an enrichment in myeloma (supposedly somatic) as compared to GnomAD (supposedly germline)? What is the evidence that TP53 P72R is somatic in your MM cohort?

R: Our study included a limited number of patients (n=51) and also, we analyzed bulk BM and enriched plasma cells (not a homogeneous distribution of samples), therefore calculating an allelic frequency and comparing it to the frequency in GnomAD Database would not be an accurate representation. Also, considering that gnomAD Database query is used mainly for germline variants, we have query it only for P72R polymorphism (TP53 gene).

We don’t claim that P72R is a germline variant. Rather otherwise, P72R variant is supposedly germline, according to ClinVar Database. The germline status is supported by our VAF in a range of 42–100%, which may suggest heterozygosity and homozygosity status, respectively.

Line 206: 0.7366 – is this a percentage or a fractional abundance?

R: In this case (according to gnomAD Database), the allele frequency is expressed fractional (allele count / number of alleles divided by overall number of alleles available for evaluation). We have updated the text: “The P72R polymorphism is most common in higher latitudes and colder climates and according to gnomAD Database, it has the highest allelic frequency in European population (0.7366)”

Line 337: What is the final evidence that NGS of CD138 is superior over NGS of bulk BM DNA? This relates to the major point stated above.

R:  Due to the major limitation of lack of sample availability, discussed at point 1, we are unable to definitively claim superiority of NGS performed on CD138 fraction vs bulk BM and we avoid strong wording that makes such claim in the manuscript. However, our finding – the overall lower abundance of genetic abnormalities, both cytogenetic and molecular, in the bulk BM compared to CD138+ enriched cells, even from different patients, is suggestive for this hypothesis and in agreement with previously published data (Haematologica. 2013 Feb;98(2):279-87).

Reviewer 2 Report

Comments and Suggestions for Authors

This is a well written nice manuscript. The issue is the clinical significance, and as disease is focal and heterogeneous single-point bone marrow sampling cannot capture the tumor heterogeneity and is difficult to repeat for serial assessments. So were there any correlations of the mutations and VAFs with any of the clinical parameters. detailed in Suppl Table 2 that are very limited. Any correlation with response to treatment. Manuscript can benefit by adding the treatments the study patients received and correlation with responses. We need a study limitation paragraph at the discussion. It will be nice to discuss the study finding in comparison to previous publications in the field ,see review at  Hematology Am Soc Hematol Educ Program (2022) 2022 (1): 349–355.Hematology Am Soc Hematol Educ Program (2022) 2022 (1): 349–355.https://doi.org/10.1182/hematology.2022000347).But mostly the authors need to discuss their results and methodology relevance to the clinic and MM field in view of the rapidly developing field of performing the NGS and mutation panel from circulating myeloma cells-the so called liquid biopsy.

Author Response

Thank you very much for your honest review. You make very good points and our manuscript can really benefit from them. I have attached the file with our answers.

Q: Manuscript can benefit by adding the treatments the study patients received and correlation with responses.

R: We have added the treatment details in the supplementary table 2. Due to the relatively small size of our cohort, definitive correlation between the presence of pathogenic variants and clinical outcomes are hard to make and would be speculative at this cohort size. Larger sample size studies could address this limitation. That being said, we identified 11/51 patients that were refractory to the last line of therapy before inclusion in the study – either refractory to induction or to maintenance therapy and we are referring to this subgroup as “refractory” in the manuscript. Of these 11 patients, 5 (45%) have at least one cytogenetic abnormality and/or somatic mutation in the MAPK pathway. These mutations co-occur with cytogenetic abnormalities. We have also created supplementary table 3 with the co-occurrence of mutations and cytogenetic lesions in this refractory group and are discussing these findings in the discussion panel.

Q: We need a study limitation paragraph at the discussion. It will be nice to discuss the study finding in comparison to previous publications in the field ,see review at  Hematology Am Soc Hematol Educ Program (2022) 2022 (1): 349–355.Hematology Am Soc Hematol Educ Program (2022) 2022 (1): 349-355.https://doi.org/10.1182/hematology.2022000347). But mostly the authors need to discuss their results and methodology relevance to the clinic and MM field in view of the rapidly developing field of performing the NGS and mutation panel from circulating myeloma cells-the so called liquid biopsy.

R: We added a study limitation paragraph in the discussion section that covers a few points – the lack of sample heterogeneity, the limited cohort size and limited clinical correlations. We cited the indicated paper and we’re discussing our findings in the light of novel other findings with clinical translational potential – use of other targets with prognostic value (MYC rearrangements), use of NGS for MRD detection and, as suggested, performing NGS from liquid biopsies versus bone marrow samples.
